# Epimutations and Their Effect on Chromatin Organization: Exciting Avenues for Cancer Treatment

**DOI:** 10.3390/cancers15010215

**Published:** 2022-12-29

**Authors:** Asad Mohammad, Sudhakar Jha

**Affiliations:** Department of Physiological Sciences, College of Veterinary Medicine, Oklahoma State University, Stillwater, OK 74078, USA

**Keywords:** epigenetics, DNA methylation, histone modification, cancer, epigenetic inhibitors

## Abstract

**Simple Summary:**

Epigenetic modifications, such as DNA methylation and histone modification, have been found to alter in various cancer types. These modifications lead to uncontrolled cellular proliferation, evasion from apoptosis, and metastasis. Deregulation in epigenetic pathways often results in the suppression of tumor-suppression genes or activation of oncogenes in cancers. Inhibitors targeting deregulated enzymes can restore balance by reactivating altered pathways. Several inhibitors that target DNA methylation and histone modifications are currently being used in clinics and have shown promising results in cancer therapeutics.

**Abstract:**

The three-dimensional architecture of genomes is complex. It is organized as fibers, loops, and domains that form high-order structures. By using different chromosome conformation techniques, the complex relationship between transcription and genome organization in the three-dimensional organization of genomes has been deciphered. Epigenetic changes, such as DNA methylation and histone modification, are the hallmark of cancers. Tumor initiation, progression, and metastasis are linked to these epigenetic modifications. Epigenetic inhibitors can reverse these altered modifications. A number of epigenetic inhibitors have been approved by FDA that target DNA methylation and histone modification. This review discusses the techniques involved in studying the three-dimensional organization of genomes, DNA methylation and histone modification, epigenetic deregulation in cancer, and epigenetic therapies targeting the tumor.

## 1. Introduction

In the Eukaryotic cell, genetic material is organized into a condensed structure inside the nucleus called chromatin. Chromatin is composed of ~147 base pairs (bp) of DNA wrapped around nucleosomes, which consist of octamers of core histone proteins H2A, H2B, H3, and H4 [1]. In the nucleus, chromatin forms higher-order structures such as fibers, loops, domains, etc. These fibers are typically 5–24 nm in diameter and folded irregularly into loops and topologically associated domains (TAD) [2,3]. These loops range in size from kilobases (Kb) to megabases (Mb) [4]. The formation of chromatin loops is associated with the activation or repression of genes [5,6]. The looping of chromatin brings upstream regulatory elements, such as enhancers and promoters, closer to neighboring genes which in turn regulate their expression. TADs are key units of the three-dimensional (3D) genome organization. TADs are formed by the physical compartmentalization of chromatin extrusion and insulation [7], which are regulated by multiple factors, including DNA methylation, histone modification, and DNA binding proteins such as CCCTC-binding factor (CTCF) and the cohesin complex. Epigenetics not only regulates gene transcription through promoters and enhancers but also genome stability. The deregulation of these processes can alter gene expression, which is often observed in cancers. These epigenetic marks are dynamic and reversible, and recent advances in the field have indicated that the enzymes involved in regulating these epigenetic marks are druggable targets.

In the following sections, we will discuss techniques involved in investigating high-order chromatin organization, factors regulating chromatin, and how mutations in these chromatin-remodeling complexes result in diseases such as cancer.

## 2. Techniques to Study Chromatin Interactions

In higher eukaryotes, chromosomes do not occupy random locations during the cell cycle. Still, they are arranged in specific patterns, where each chromosome occupies a defined and mutually exclusive region called chromosome territory (CT) [8]. Dekker et al. [9] developed an approach based on molecular biology to study chromatin organization in a 3D space called chromosome conformation capture (3C). 3C technique allows us to study the 3D organization of genomes and has revealed an intricate relationship between transcription and genome organization. As a result, a number of derivative methods have been developed [10], including chromosome conformation capture on a chip (4C), chromosome conformation capture carbon copy (5C), Hi-C, Micro-C, Capture-C, and the chromatin interaction analysis paired-end tag sequencing (ChIA-PET) allowing us to infer 3D chromosome organization.

### 2.1. Chromosome Conformation Capture (3C)

To determine genomic architecture, 3C quantifies the frequency of contacts between two distal selected genomic elements, such as enhancers and promoters, in a population of cells. In 3C, cells are formaldehyde cross-linked, chromatin is isolated and treated with a restriction enzyme and followed by the ligation of DNA fragments performed at low DNA concentrations, which aids intramolecular ligation of cross-linked chromatin segments (Figure 1a). At the point of cross-linking, a one-dimensional template representing a cellular 3D environment is captured. This template is used for semiquantitative or quantitative polymerase chain reaction (PCR) to interrogate short- and long-range chromatin interactions at the region of interest (Figure 1b) [11]. Despite 3C’s ability to visualize the genome at high resolution, there are still some limitations, including the need for genomic information that can be used to amplify regions of interest. In addition, 3C can only detect contact within a few hundred Kb.

### 2.2. Chromosome Conformation Capture-on-Chip (4C)

Using 4C, a region of interest can be identified at the genome-wide level. To analyze the chromatin organization, 3C methodology is used in the initial steps of the protocol (Figure 1a), followed by a ligation step after obtaining the 3C template (Figure 1c), resulting in small DNA circles. A reverse PCR strategy is then employed to amplify the contacting sequences using primers designed on the region of interest, also known as viewpoint. Using next-generation sequencing (NGS), one of the most common sequencing methods, viewpoint-contacting regions are identified. 4C can identify all genomic regions involved in the gene- or regulatory interaction [12].

### 2.3. Chromosome Conformation Capture Carbon Copy (5C)

Using 3C and 4C methods, it is not possible to infer genome organization. However, 5C provides information about multiple fragments establishing contacts in a large genomic region and can be investigated in parallel [13,14]. Based on the conventional 3C protocol, it uses highly multiplexed ligation-mediated amplification with pools of locus-specific primers to interrogate the 3C library, followed by sequencing or microarrays to quantify ligated pairs (Figure 1d). 5C enables the analysis of all interactions between two sets of genomic loci-one set recognized by the forward primers, the other by reverse primers. With thousands of 5C primers, 5C can be multiplexed very efficiently.

### 2.4. Hi-C

A series of “all versus all” methods emerged with the development of high-throughput sequencing technology. Hi-C has been extensively utilized to detect altered conformational changes in human diseases such as cancer. Hi-C method generates genome-wide contact maps without requiring specific primers [15,16]. Hi-C follows the same initial step as 3C to generate a template (Figure 1a). The sticky ends are filled in with biotin-labeled nucleotides and blunt-end ligated after restriction enzyme digestion. For high-throughput sequencing, only biotinylated junctions are selected by purifying them with streptavidin beads (Figure 1e).

### 2.5. Micro-C

Micro-C is a 3C-based method that is an enhanced and new variation of the Hi-C [17]. Micro-C was developed to elucidate the full spectrum of chromatin folding that can be observed from nucleosomes to chromosomes. Furthermore, Micro-C can provide more information on the boundaries of chromatin domains and identify new chromatin interactions and loops, providing higher resolution and higher signal-to-noise ratio. Micrococcal nuclease (MNase) is used in this protocol to digest the genome (Figure 1f), which results in the fragmentation of the chromatin into mononucleosomes. The protocol can overcome the resolution limitation of restriction enzyme-based approaches, thereby enabling a more accurate estimation of nucleosomes’ proximity to one another.

### 2.6. Chromatin Interaction Analysis Paired-End Tag Sequencing (ChIA-PET)

ChIA-PET is an innovative method for analyzing long-range chromatin interaction [18]. ChIA-PET is a combination of chromatin immunoprecipitation (ChIP) and 3C-type analysis. This method detects long-range chromatin interactions associated with a specific protein at a genome-wide level. ChIA-PET involves cross-linking DNA and proteins, fragmenting them by sonication, and capturing them with protein-specific antibodies. Biotin-labeled oligonucleotides with MmeI restriction sites are used to link the captured chromatin to the biotin-labeled oligonucleotides. After ligating the adjacent linkers, DNA fragments with paired-end tags (PETs) are obtained by digesting the linkers with MmeI restriction enzyme, followed by high-throughput sequencing using PETs (Figure 1g).

### 2.7. Capture-C

Capture-C is an improved version of 3C-sequencing and Hi-C and is a “many vs. all” approach. Capture-C is based on a standard 3C experiment, but the fragments ligated in the 3C are sonicated at the end of the experiment (Figure 1h). Capture-C uses biotinylated probes to capture target regions before they are sequenced downstream with high-throughput sequencing [19]. In Capture-C, the selected regions of interest undergo enrichment before sequencing. As a result, there is an increase in throughput and an improvement in genome resolution. This approach was later applied to Hi-C libraries; as a result, Capture Hi-C (CHi-C) was developed [20]. An improvement to Capture-C is called next-generation (NG) capture-C, which uses biotinylated single-stranded DNA (ssDNA) oligonucleotides in sequential “double capture” for indexed multiplexed 3C libraries [21].

Inside the nucleus, chromosomes are folded into a 3D structure. Genomic architecture has been studied using a variety of techniques. Among them, 3C-based techniques have become one of the most effective ways to understand how spatial genome organization is established and maintained. 3C technologies have helped our understanding of how changes in the genome result in a disease such as cancer [10].

## 3. Epigenetic Factors Involved in Regulating Chromatin Organization

This section will discuss proteins involved in modifying chromatin and how they regulate transcriptional landscape.

### 3.1. DNA Methylation

DNA methylation involves a covalent transfer of methyl group to cytosine in position 5, resulting in 5-methylcytosine (5 mC); this process is catalyzed by DNA methyltransferases (DNMTs) [22]. It is an essential epigenetic modification involved in regulating gene expression, cellular proliferation, differentiation, and stem cell maintenance [23]. The methylation pattern of DNA is established and maintained by DNMTs [22]. There are three major types of DNMTs; DNMT1, DNMT3a, and DNMT3b. DNMT3a and DNMT3b are involved in de novo DNA methylation, and DNMT1 maintains the methylation of DNA during DNA replication [24,25]. Most DNA methylation occurs on cytosines preceding guanines, or CpG sites, creating a transcriptionally repressive chromatin state. As a result of DNA methylation, a few transcription factors are unable to bind to DNA, and proteins that recognize methylated DNA through their methyl-CpG binding domain (MBD) are recruited to inhibit gene expression. In normal cells, CpG islands in gene promoters are usually unmethylated and are transcriptionally permissive, which keeps the chromatin open and enhances transcription. CpG islands-containing tumor suppressor genes at the promoters are methylated in tumor cells, turning euchromatin into heterochromatin, while in normal tissues, these CpG islands are typically unmethylated [26].

### 3.2. Histone Acetylation

Histone acetylation involves adding an acetyl group to the lysine residues of histones. All four core histone proteins can be acetylated in the nucleosome octamer at various lysine residues [27]. Acetylated histone tail facilitates nucleosome assembly and creates a transcriptionally active chromatin environment, allowing DNA sequences to be exposed and transcribed.

Histone acetylation levels are maintained by two enzymes, histone acetyltransferases (HATs) and histone deacetylases (HDACs) [28]. HATs utilize acetyl CoA as a cofactor to transfer an acetyl group to the ε-amino group of the lysine side chain of histone and non-histone proteins. HATs are classified into three major groups: Gcn5-related N-acetyltransferases (GNATs), MYSTs (MOZ, Ybf2, Sas2, TIP60), and orphans (CBP/EP300) [29]. Depending upon the acetylation of histone in the cytoplasm or the nucleus, HATs are categorized into two classes, type-A, and type-B. Nucleosomes and proteins involved in access to DNA are regulated by type-A HATs, whereas type-B HATs acetylate free histone, predominantly in the cytoplasm. HATs act as tumor suppressors and oncogenes, suggesting acetylation is critical in maintaining the balance. Deregulation of this balance often results in diseases such as cancers [30,31].

HDACs remove acetylation from histone tails and non-histone proteins. Human HDACs can be divided into two families based on their catalytic mechanisms. Zn^2+^ dependent enzymes (Class I, II, and IV) and NAD^+^ cofactor-dependent enzymes (Class III) [32]. There are 18 human HDACs which are further categorized into four classes. Class I (HDAC1, HDAC2, HDAC3 and HDAC8), Class IIa (HDAC4, HDAC5, HDAC7 and HDAC9), Class IIb (HDAC6 and HDAC10), Class III (Sirtuins 1–7), and Class IV (HDAC11) [33].

### 3.3. Histone Methylation

Histone methylation is one of the crucial post-transcriptional modifications. The methylation of histones does not affect histone-DNA binding directly but instead interferes with chromatin-binding factors’ interaction with DNA. To activate or repress gene expression, histone lysine residues can be mono-, di-, or tri-methylated (me1, me2, or me3) [34]. Methylation of histones has a diverse outcome; for example, methylation on H3K4, H3K36, and H3K79 are activating marks and are in an actively transcribed region, whereas H3K9, H3K27, and H4K20 are usually associated with silenced gene expression or condensed chromatin [35].

The methylation of histones is controlled by two families of enzymes: histone methyltransferases (HMTs) and histone demethylases (HDMs). HMTs transfer a methyl group to the lysine and arginine residues of the target protein, whereas HDMs can remove the methyl group [36]. This addition can be one, two, or three methyl groups of S-adenosyl-L-methionine transferred to lysine or arginine residues [37,38]. There are two families of histone demethylases: amino oxidase homolog lysine demethylase 1 (KDM1) and Jumonji C (JmjC) domain-containing histone demethylases [39]. The KDM1 family consists of two members, KDM1A, also known as lysine-specific demethylase 1 (LSD1), and KDM1B. KDM1A removes mono- and di-methylated lysine 4 or lysine 9 of histone H3 [40]. KDM1B is the other amine oxidase homolog that targets H3K4me1 and H3K4me2 [41]. A second group consists of demethylases containing the JmjC domain. Approximately 20 of the identified JmjC domain proteins demethylate lysines [42]. Unlike KDM1, these enzymes remove trimethylations [43].

### 3.4. Histone Phosphorylation

Phosphorylation of cellular proteins is one of the common post-translational modifications (PTM). Interestingly, histone proteins are also phosphorylated on serine, threonine, and tyrosine residues, resulting in changes in chromatin organization [37]. The phosphorylation of histone tails occurs predominantly at the N-terminal amino acid of histone tails. Phosphorylation of histone tails reduces the positive charge on the histones and weakens histone-DNA interaction, which leads to DNA access by protein complexes [44,45]. Several biological processes depend on core histone phosphorylation, including transcription, cell cycle regulation, DNA repair, and apoptosis [46]. Amongst the intracellular signaling pathways involved in chromatin remodeling, phosphorylation plays an important role in connecting the signaling pathways and chromatin [47]. For example, phosphorylation of histone H3 on serine 10 (H3S10) by Aurora-B dissociates heterochromatin protein 1 (HP1) from chromatin and prevents heterochromatin formation [48]. Some pathological events are associated with abnormal kinase activity, cancer being one of the most prominent [49,50,51].

### 3.5. Ubiquitination

Ubiquitin is an evolutionarily conserved protein among eukaryotes. Ubiquitin is a 76 amino acid residue polypeptide. It has a molecular mass of 8.5 kDa, and the ubiquitin exists as a free molecule or covalently conjugated with other proteins [52]. In cells, ubiquitin mostly functions as part of a multistep proteolytic pathway. Ubiquitylation of protein involves three main enzymes, ubiquitin-activating enzymes (E1), binding enzymes (E2), and ligases (E3) [53]. In cells, ubiquitylation is usually associated with the degradation of proteins. However, the addition of ubiquitin to histones acts as a regulatory mechanism [54,55]. In fact, in the nucleus, histones are the most abundant ubiquitinated proteins [56], and their ubiquitination is crucial for transcription, maintaining chromatin structure, and repairing DNA. Ubiquitination also plays a significant role in regulating tumor-suppressing and tumor-promoting pathways [57]. For example, *TP53*, a tumor suppressor gene that prevents instability of the genome, is targeted for degradation by ubiquitylation. By interacting with the N-terminus transactivation domain (TAC) of p53, MDM2 ubiquitinates p53 and triggers degradation by the ubiquitin-proteasomes system [58]. Another tumor suppressor gene, *BRCA1*, which is involved in DNA damage repair and the control of cell cycle checkpoints, is known to act as an E3 ligase, and mutations in E3 ubiquitin ligase activity result in tumorigenesis [59].

### 3.6. SUMOylation

Small ubiquitin-like modified proteins (SUMO) bear many similarities to ubiquitin proteins. SUMO is an evolutionarily conserved protein expressed in all eukaryotes [60]. Like ubiquitin, SUMO proteins are approximately 10 kDa in size. SUMOylation is an important post-translational protein modification that is conjugated to lysine residues of the target proteins. Four *SUMO* isoforms have been identified, which are *SUMO1*, *SUMO2/3* and *SUMO4* [61]. The SUMO proteins are initially translated as C-terminally stretched precursors, which are then cleaved by SUMO-specific proteases into proteins ending in a pair of glycine residues. The alpha-carboxylates at the C-terminus are used to attach other proteins covalently, and this process is known as sumoylation [62]. Target proteins of SUMO are often transcriptional coactivators or corepressors [63]. SUMO protein is primarily found in the nucleus, and diverse cellular processes are regulated by sumoylation, including transcription, DNA replication, cell-cycle progression, mitochondrial dynamics, DNA repair, and apoptosis [64,65].

### 3.7. Citrullination

Citrullination is another PTM catalyzed by peptidylarginine deiminase (PAD). Unlike epigenetic modifications, citrullination involves a Ca^2+^-driven enzymatic conversion of arginine residues to citrulline [66]. There are five isoenzymes, PAD1, PAD2, PAD3, PAD4, and PAD6 identified in humans. PADs have been shown to citrullinate histone H1, H2A, H3, and H4 [67,68,69]. It has been reported that calcium-activated PAD4 reduced the methylation of recombinant histones H3 and H4 and can affect chromatin structure by promoting decondensation in vivo [68]. Interestingly, citrullinated histones have been shown to interact with other modified histones, such as those that are methylated. In its dual enzyme activity, PAD4 converts arginine and methylated arginine into citrulline through deamination and demethylation, respectively [68,70], suggesting a crosstalk between methylation and citrullination. Citrullination of histones plays a significant role in transcription, pluripotency [69], development of embryos [71] and neutrophil extracellular traps (NET) formation [72].

### 3.8. ADP Ribosylation

ADP-ribosylation is a covalent post-translational modification of proteins catalyzed by ADP-ribosyltransferases. ADP-ribosylation is a process in which mono or poly-ADP-ribose molecules are transferred from the cofactor nicotinamide adenine dinucleotide (NAD) to the target protein by a process known as mono-ADP-ribosylation (MARylation) or poly-ADP-ribosylation (PARylation). In addition to the core histones, H1 linker histones are ADP-ribosylated. ADP-ribosylation occurs in glutamic acid, aspartic acid, lysine, arginine, and serine [73,74,75]. A majority of histone ribosylation events in mammals are catalyzed by PARP1, which is supported by PARP2/3/7/10 [76]. It plays an essential role in cell cycle regulation, DNA damage response [77], replication, and transcription [78].

### 3.9. Histone Glycosylation

Proteins and lipids undergo glycosylation, the process in which monosaccharides are attached to proteins and lipids to modify their properties. Histone glycosylation is a post-translational modification. In the human genome, the modification of the hydroxyl group of a serine or threonine residue by monosaccharides on the core histones proteins H2A, H2B, H3, and H4 results in conjugation to O-linked N-acetylglucosamine (O-GlcNAc). It has been reported that various core histone proteins have O-GlcNAc modifications with varying functions, for example, H2AT101, H2BS36, H3S10, and H4S47, as well as H3T32 [79,80]. Glycosylation has been shown to regulate the cell cycle, transcriptional activation, and chromatin dynamics [81,82,83,84]. A wide variety of cancer types have been found to have aberrant O-GlcNAcylation, and studies suggest that O-GlcNAcylation may play a regulatory role in cancer development [85].

### 3.10. Proline Isomerization

In amino acids, functional groups on carbon chains are positioned according to their isomeric form: cis and trans. Protein folding is affected by the changes in proline residues between the cis and trans forms. Peptidylprolyl isomerases (PPIases), which operate by switching between conformations, are enzymes that facilitate the isomerization process. Initially, Tony Kouzarides’ lab reported proline isomerization on the tail of histone H3 [86]. Histone H3P38’s trans-proline conformation is essential for methylating lysine 36 of histone H3 (H3K36). On the other hand, cis-isomerization is related to lower H3K36 methylation and decreased transcription elongation due to cis-isomerization. The acetylation of H3K14 facilitates H3P16 trans-isomerization. As a result, H3K4me3 levels are reduced in vivo, indicating that H3P16 trans-isomerization may be involved in the repression of transcription [87].

## 4. Epimutations, TADs and Cancer

### 4.1. Writes, Readers, and Erasers

Numerous cancers have been linked to epigenetic modifications, such as DNA methylation and histone modifications [88,89]. These enzymes can further be classified as writers (DNA methyltransferases [DNMTs], histone acetyltransferases [HATs] and histone methyltransferase [HMTs], etc.), erasers (histone demethylases [HDMs], and histone deacetylases [HDACs]), readers (bromodomain and chromodomain proteins that are known to read acetylated and methylated residues, respectively) (Table 1 and Figure 2), and insulators (CTCF and the cohesin complex).

### 4.2. Loops, TADs, and Insulators

The DNA is wrapped around histones to form chromatin which further undergoes high-order organization resulting in loops and TADs (Figure 3). This high-order organization is facilitated by two key protein complexes, CTCF and cohesin. CTCF is a highly conserved, ubiquitously expressed, and essential protein in eukaryotes [93]. CTCF is an 82-kDa protein that comprises three separate regions, an 11-zinc fingers region with a central zinc-finger domain that binds to the DNA flanked by large C- and N-terminal domains [94]. CTCF acts as both a transcriptional activator and repressor by interacting with cofactors involved in transcriptional processes [95]. CTCF functions as an insulator when positioned between the promoter and enhancer through its ability to form loops and domains [96]. Cohesin is a ring-shaped protein complex composed of four core subunits encoded by the genes *Smc1*, *Smc3*, *Rad21*, and *STAG1/2* [97]. Cohesin and CTCF colocalize in the mammalian genomes to form high-order chromatin structures [98]. One of the functions of cohesin is to fold chromatin, create CTCF loops, and bring cis-acting elements, such as enhancers, to gene promoters in proximity [99]. By doing so, the cohesin protein complex plays an important role in cell cycle regulation, DNA repair, chromatin organization, and gene expression regulation [100].

TADs are separated by insulators that maintain higher-order chromatin structures. Changes in their expression level or mutation in the binding site result in an abnormal scenario. Indeed, mutations in *CTCF* and cohesin have been identified in breast, prostate, and uterine cancers [101]. In prostate cancer, loss of *CTCF* results in hypermethylation at CTCF binding sites. The cohesin subunits (*Smc1*, *Smc3*, *Rad21*, and *STAG1/2*) are also frequently mutated in cancer. For example, *RAD21* somatic mutation and amplification are found in both solid and hematopoietic tumors. Somatic mutation is mainly found in hematopoietic malignancy, while overexpression is associated with solid tumors [102]. Other components of the cohesin complex are also known to be mutated in cancers. For example, *SMC1A* is frequently mutated in colorectal cancer, and its variants have been detected in leukemia and bladder cancer [103]. Studies have shown that *STAG2* acts as a tumor suppressor gene. *STAG2* mutations are mostly nonsense, frameshift, and splice mutants found in melanoma, acute myeloid leukemia (AML), Ewing’s sarcoma, and myelodysplastic syndrome (MDS) [104].

### 4.3. Epigenetic Inhibitors

Epigenetic mechanisms account for one-third to one-half of all genetic alterations in cancer [105]. Deregulation of their enzyme activity can be targeted by small molecules such as DNMT inhibitors (DNMTi), HDAC inhibitors (HDACi), DOT1L inhibitors, EZH2 inhibitors, and BET inhibitors [106,107] (Table 2). In this section, we will discuss a few of the inhibitors showing promising preclinical and clinical data for cancer treatment.

#### 4.3.1. DNA Modification Inhibitors

As discussed earlier, malignant transformation is frequently associated with hypermethylation of CpG-rich regions leading to the silencing of tumor suppressor genes, and this can be prevented by exposing cancer cells to DNMTi. DNMTi are potent therapeutic agents for cancer treatment as they reverse the hypermethylation of DNA in tumor cells. DNMTi have been classified into two types; nucleoside analogs and non-nucleoside analogs. Nucleoside analogs are made from modified cytosine rings with nitrogen at position 5 (instead of carbon), and these modified analogs are incorporated into newly synthesized DNA. In contrast, non-nucleoside DNMTi are small molecules directly targeting the catalytic sites instead of being incorporated into DNA [163]. Azacytidine (5-azacytidine) and decitabine (5-aza-2′-deoxycytidine) are the two well-studied nucleoside analogs and potent DNA hypomethylating agents. Mechanistically, azacytidine and decitabine form an irreversible complex with DNMTs, causing degradation of the complex [164]. The FDA has approved these compounds for treating AML and MDS [107] as they promote differentiation and reexpression of inactivated genes. Decitabine and azacytidine are also being studied as therapeutic options in multiple solid cancers [165,166]. DNA methylation is inhibited in a dose-dependent manner by decitabine. Interestingly, when used at low doses, decitabine can reactivate silenced genes, but at high doses, it becomes cytotoxic, and at all doses, myelosuppression is a prominent side effect [167]. Second-generation DNMTi, such as SGI110, were demonstrated to be more stable and less toxic than first-generation DNMTi [168,169].

Demethylation of DNA is mediated by the ten-eleven translocation (TET) protein family (TET1/2/3), which converts 5-methylcytosine (5mC) into 5-hydroxymethylcytosine (5hmC) during DNA demethylation [170]. Isocitrate dehydrogenase (IDH) is responsible for generating the metabolite (2-oxoglutarate), which is necessary for the activity of the TET enzyme. AG-120 (ivosidenib) and AG-221 (enasidenib) are IDH inhibitors that ablate IDH1/2-mediated conversion of α-ketoglutarate to 2-hydroxygluterate and accumulation of 2-hydroxygluterate inhibits TET activity [171]. It has been demonstrated that targeting DNMTs not only activates tumor suppressor genes but also restores optimal methylation levels, which then modulates the binding of proteins such as CTCF and cohesin. For instance, in breast and prostate cancer, loss of *CTCF* copy number leads to hypermethylation of DNA at surrounding CTCF binding sites [172]. Inhibition by azacytidine reverses the change in gene expression after the loss of CTCF sites [173].

#### 4.3.2. Histone Modification Inhibitors

A number of epi-drug that inhibit histone modification have been developed and used for cancer treatment. For example, HDACi such as vorinostat (SAHA), belinostat (PXD-101), panobinostat (LBH589), and romidepsin (FK228) have shown promising results in treating cancer [174,175]. For example, exposure to vorinostat induced cell cycle arrest in refractory/relapsed B-cell lymphoma and is due to an increase of *p21* and the acetylation of histone H3 [176,177]. Similarly, another HDACi, belinostat, has been approved for use in relapsed or refractory peripheral T-cell lymphoma (PTCL), panobinostat has been used for multiple myeloma (MM) [138,178], and romidepsin has been approved for the treatment of cutaneous T-cell lymphoma (CTCL) [179]. Like HDACi, HAT inhibitors (HATi) have also shown promising results. For example, the small molecule C646 selectively inhibits p300/CBP activity [180,181] and induces cell cycle arrest and apoptosis. In addition to HATi and HDACi, HMT inhibitors (HMTi) have also been developed as anticancer drugs. Pharmacological inhibition of DOT1L by EPZ-5676 reduces H3K79me methylation and decreases proliferation. Loss of cohesin resulted in increased self-renewal and increased *HoxA9* expression, which is reversed by DOT1L inhibition (pinometostat) [182]. Inhibition of DOT1L in cohesin-depleted cells also shows a decrease in H3K79me2 and a concomitant increase in H3K27me3 [182]. Deregulation in another HMT, *LSD1*, promotes acute lymphoblastic leukemia (ALL) and AML [183,184]. Several LSD1 inhibitors (TCP, ORY-1001, GSK-2879552, and ORY-2001) are currently being evaluated for cancer treatment and have been shown to inhibit proliferation, differentiation, and invasion in both in vitro and in vivo studies [185].

## 5. Future Perspective

The human genome is organized in a complex 3D structure. As epigenetic mechanisms play a significant role in regulating tumorigenesis, understanding how DNA and histone modifications are involved in maintaining genome organization will be an exciting avenue in the future. Furthermore, since epigenetic pathways are reversible, targeting the enzymes deregulated in cancer can restore this balance, offering promising avenues for anticancer treatment. Clustered regularly interspaced short palindromic repeats (CRISPR)—CRISPR-associated endonuclease (Cas9) has emerged as a powerful genome-editing technique that allows specific changes to be made to the genome with high efficiency and specificity. CRISPR-Cas9 genome editing is being explored extensively for cancer therapy and has been used to target epigenetic modification. Several epigenetic therapies reverse gene expression, such as epi-drugs and combinatorial treatment. However, a significant obstacle to their clinical application is that epi-drugs might activate off-target genes in normal cells that cause mutagenesis. Since the CRISPR-Cas9 system enables specificity, modified versions such as CRISPR activation (CRISPRa) and CRISPR inactivation (CRSPRi) might be effective tools for cancer treatment.

## Figures and Tables

**Figure 1 cancers-15-00215-f001:**
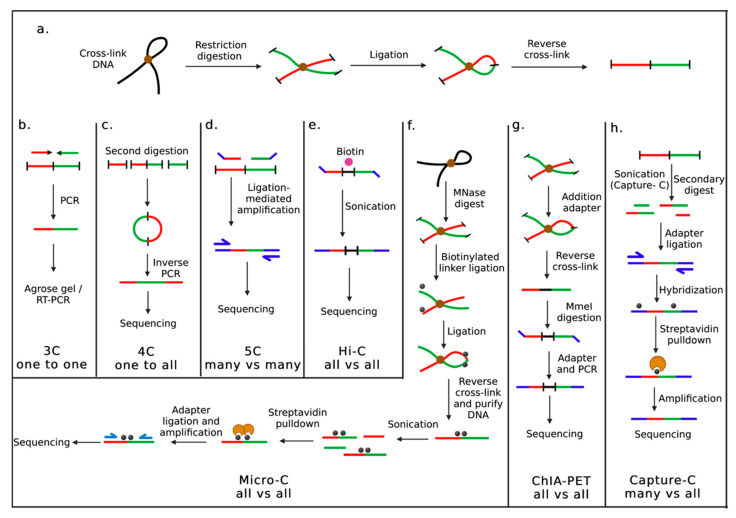
Techniques for studying chromatin organization in three-dimensional (3D) space. Steps involved in studying the 3D organization of genomes, (**a**) A common step in 3C-related techniques is crosslinking chromosomes with formaldehyde and their fragmentation by restriction digestion. The organization of chromatin can be visualized in 3D space using several different detection approaches such as (**b**) 3C (chromosome conformation capture), (**c**) 4C (chromosome conformation capture-on-Chip), (**d**) 5C (chromosome conformation capture carbon copy), (**e**) Hi-C, (**f**) Micro-C, (**g**) ChIA-PET (chromatin interaction analysis paired-end tag sequencing) and (**h**) Capture-C. This figure was created using BioRender.com access on 14 December 2022.

**Figure 2 cancers-15-00215-f002:**
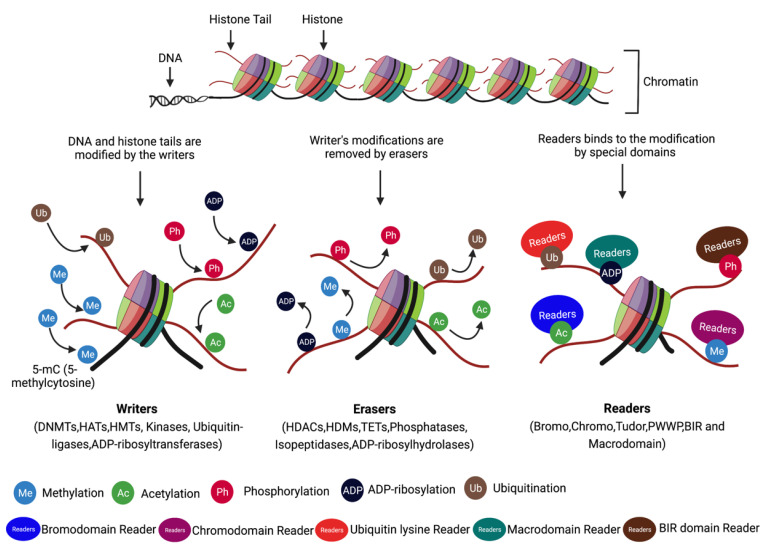
Factors involved in regulating epigenetic code. Histones undergoes different PTMs such as methylation, acetylation, phosphorylation, ubiquitination and ADP-ribosylation. Writers and erasers are involved in adding and removing PTM, respectively. Readers with specific domains recognizes the PTMs. This figure was created using BioRender.com (accessed on 14 December 2022).

**Figure 3 cancers-15-00215-f003:**
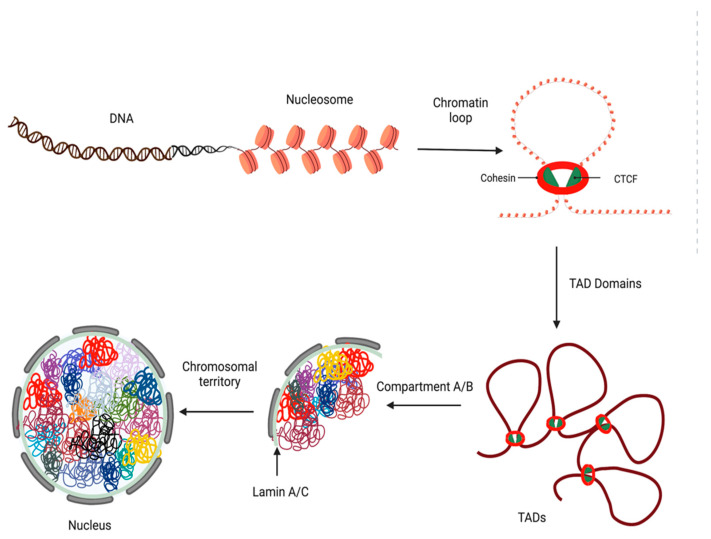
Several levels of organization are present in genomes. Inside the nucleus, DNA (black) is wrapped around nucleosomes (orange), which consist of octamers of core histone proteins H2A, H2B, H3, and H4. Cohesin (red) and CTCF (green) colocalize in the mammalian genomes to form high-order chromatin structures. Chromatin fibers are folded into loops, bringing upstream regulatory elements close to gene promoters. TADs (brown) are formed when chromatin is extruded and insulated in a physical compartment. DNA resides inside chromosome territories (multiple colors), generating non-random arrangements of chromosomes and genes within the nucleus. This figure was created using BioRender.com (accessed on 14 December 2022).

**Table 1 cancers-15-00215-t001:** Proteins involved in chromatin organization, adapted from [90,91,92].

Modification	Gene/Components	Enzyme/Action
Writers	*DNMT1*	DNA methyltransferase
	*DNMT3a*	DNA methyltransferase
	*DNMT3b*	DNA methyltransferase
	*EZH1/2*	Histone methyltransferase
	*DOT1L/KMT4*	Histone methyltransferase
	*EP300 (P300/KAT3B)*	Histone acetyltransferase
	*CREBBP (CBP/KAT3A)*	Histone acetyltransferase
	GNAT family *(GCN5)*	Histone acetyltransferase
	MYST family (*TIP60/KAT5*)	Histone acetyltransferase
	*EHMT1/2*	Histone methyltransferase
	*SUV39H1/2*	Histone methyltransferase
	*PRMT1/2/4/5/6/7*	Histone methyltransferase
	*KMT7*	Histone methyltransferase
	AuroraB (*AURKB*)	Kinases
	WSTF (*BAZ1B*)	Kinases
	BRCA1-BARD1	Ubiquitin-ligases
	*PARP1*	ADP-ribosyltransferases
Erasers	*TET1/TET2*	DNA demethylation
	*HDAC 1-3,8* (Class I)	Histone deacetylase
	*HDAC 4,5,7,9* (Class IIa)	Histone deacetylase
	*HDAC 6,10* (Class IIb)	Histone deacetylase
	HDAC Class III: *SIRT 1-7*	Histone deacetylase
	*HDAC 11* (Class IV)	Histone deacetylase
	*KDM1A/LSD1, KDM1B/LSD2*	Histone demethylase
	*KDM2A/2B, KDM3A/3B*	Histone demethylase
	*KDM4A/4B/4C/4D/4E*	Histone demethylase
	*KDM5A/5B/5C/5D*	Histone demethylase
	*KDM6A/6B/7/8*	Histone demethylase
	PP1 (*PPP1CA*)/PP2 (*PPP2CA*)	Phosphatases
	PPgamma (*PPARG*)	Phosphatases
	*EYA1/3*	Phosphatases
	*OTUB1/2, BRCC36, USP3/16/26/44*	Isopeptidases
	*PARG, MDO1/2, TARG*	ADP-ribosylhydrolases
Readers	*MORF, MRG15* (chromodomain)	Reads methylation
	*MBT, PHF1/19, TDRD7* (Tudor domain)	Reads methylation
	*BRPF1, NSD1-3* (PWWP domain)	Reads methylation
	*G9a/GLP* (Ankyrin repeats)	Reads methylation
	*BRD2/3/4/T* (bromodomain)	Reads acetylation
	*XRCC1, NBS1, BARD1* (BIR domain)	Reads Phosphorylation
	*14-3-3β/γ/η/ε/μ*(14-3-3 proteins BRCT domain)	Reads Phosphorylation
	*53BP1*	Ubiquitin lysine reader
	*RNF146* (Macrodomains)	Reads ADP-ribosylation
	*APLF*, *CHFR* (PBZ)	Reads ADP-ribosylation

**Table 2 cancers-15-00215-t002:** Epigenetic inhibitors in different cancer types (Adapted from [108]).

Category	Epigenetic Regulation	Target	Cancer Types	Known Compound	Reference/https://clinicaltrials.gov Identifier
Writers	DNMT	DNMT1	MDS, AML	5-aza-2′-deoxycytidine (5-aza-CdR; decitabine, Dacogen^®^)	[109]
		DNMT1	MDS, AML	5-azacytidine (5-aza-CR; Aza; Vidaza^®^)	[110]
		DNMT1, DNMT3,	Hematological malignancies, and solid tumors	Zebularine (NSC309132; 4-deoxyuridine)	[111]
		DNMT1	Hematological malignancies and solid tumors	Guadecitabine (SGI-110)	NCT01261312, NCT01752933
	HMT	DOT1L	Hematological malignancies	EPZ00477	[112]
		DOT1L	Hematological malignancies	Pinometostat (EPZ-5676)	[113]
		DOT1L	Leukemia	SGC0946	[114]
		EZH1	DLBCL	UNC1999	[115]
		EZH2	AML	3-Deazaneplanocin A (DZnep)	[116]
		EZH2	B-cell lymphoma	GSK126	[117]
		EZH2	B-cell lymphoma	Tazemetostat (EPZ-6348)	[118]
		EZH2	DLBCL	EI1	[119]
		EZH2	Non-Hodgkin lymphoma	EPZ005687	[120]
		EZH2	Ovarian cancer	GSK343	[117]
		EZH2	Breast, colon, prostate cancer	DZNep	[116]
		G9a/EHMT2	Leukemia, bladder cancer	BIX-01294	[121]
		G9a/EHMT2	Pancreatic cancer	BRD4770	[122]
		G9a/EHMT2	AML, breast cancer	UNC0638	[123]
		SUV39H1	Lymphomas	Chaetocin	[124]
		PRMT5	Solid tumors, non-Hodgkin lymphoma	GSK3326595	NCT02783300
		PRMT1	AML	AMI-408	[125]
		PRMT1, 3, 4, 6, 8	TBD	MS023	[126]
	HAT	p300	Prostate cancer	C646	[127]
		p300, CBP	MM, breast cancer, pancreatic cancer	Curcumin	[128]
Erasers	HDAC	Class I, II, IV	Leukemia, colorectal cancer, prostate cancer, and other solid tumors	Sulforaphane (SFN)	[129]
		Class I	Leukemia, colorectal cancer	Domatinostat (4SC-202)	[130]
		Class I, II, IV	Leukemia, colorectal cancer, head and neck cancer, hepatocellular carcinoma	Resminostat (4SC-201, RAS2410)	[131]
		Class I, II, IV	CTCL, Hodgkin’s lymphoma, breast cancer, head and neck cancer, prostate cancer, colorectal cancer, thyroid cancer	Panobinostat (LBH589)	[132]
		Class I, II, IV	CTCL, leukemia, prostate cancer, bladder cancer, breast cancer	Vorinostat (SAHA, Zolinza^®^)	[133]
		Class I, II, IV	CTCL	Romidepsin (depsipeptide, FK228)	[134]
		Class I (HDAC1, 9, 11)	Hodgkin lymphoma, kidney cancer, breast cancer	Entinostat (MS-275, SNDX-275)	[135]
		Class I, IV	Follicular lymphoma, Hodgkin’s lymphoma and AML, CLL, MDS, solid tumors	Mocetinostat (MGCD0103)	[136]
		Class I, II, IV	MDS, AML	Pracinostat (SB939)	[137]
		Class I, II, IV	Leukemia, colorectal cancer, lung cancer, pancreatic cancer	Belinostat (PXD101)	[138]
		HDAC-LSD1	Hematological malignancies	HDAC-LSD1 4SC-202	[139]
		Class I, II	Clinical trials: Hodgkin lymphoma, non-Hodgkin lymphoma, CLL	Abexinostat	[140]
		Class I, II, IV	Clinical trials: solid tumors	CG200745	[141]
		Class I	Clinical trials: solid tumors	CHR-3996	[142]
		Class I, II	Clinical trials: SCC	CUDC-101	[143]
		Class I, II	Clinical trials: MM, lymphoma, solid tumors	CUDC-907	[144]
		Class I, II	Clinical trials: CLL, MM, Hodgkin lymphoma	Givinostat	[145,146]
		Class I, II (HDAC1, 2, 6)	Clinical trials: solid tumors	MPT0E028	[147]
		Class I, II	Clinical trials: solid tumors, lymphoma, CTCL	Quisinostat	[148]
		Class I, II (HDAC1, 2, 3, 10)	Clinical trials: breast cancer, NSCLC	Chidamide	[149]
		Class II (HDAC6)	Clinical trials: MM, lymphoma	Ricolinostat	[150]
		Class I	Clinical trials: MM, lung cancer, pancreatic cancer	Tacedinaline	[151]
		Class I (HDAC1, 2)	Approved: CTCL, PTCL	Romidepsin	[134]
		Class I, II	Clinical trials: AML, MM	AR-42	[152]
		Class I, II	Clinical trials: solid tumors, hematological malignancies	Phenylbutyrate	[153]
		Class I, II	Clinical trials: NSCLC, MM, CLL	Pivanex	[154]
		Class I, II	Clinical trials: solid tumors, hematological malignancies	Valproic acid	[155]
	HDM	LSD1	AML, small cell lung cancer	GSK2879552	[156]
		LSD1	AML	GSK354, GSK690	[157]
		LSD1	MDS	NCD25, NCD38	[158]
		LSD1	Acute leukemia	ORY-1001	[156]
		LSD1	MDS, AML	Tranylcypromine	[156]
		HDAC-LSD1	Hematological malignancies	4SC-202	[139]
		JmjC domain proteins	TBD	GSK-J1, GSK-J4	[159]
		KDM5B	Hematological malignancies and solid tumors	EPT-103182	[160]
Readers	BET	BET proteins	Prostate cancer, AML with mixed lineage leukemia translocations, MM, NUT midline carcinoma	JQ1	[161]
		BET proteins	Clinical trials: Hematological malignancies, NUT midline carcinoma, solid tumors	I-BET762	NCT01943851, NCT01587703, [162]
		BRD2, 3, 4	Clinical trials: AML	OTX015	NCT01713582

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
