# Peer review of "Epimutations and Their Effect on Chromatin Organization: Exciting Avenues for Cancer Treatment"

_cancers, 2022, doi:10.3390/cancers15010215_

Round 1
Reviewer 1 Report
The review titled “Epimutations and their effect on chromatin organization: exciting avenues for cancer treatment” of Mohammad A. and Jha S. describes, firstly, the evolution of the techniques that has helped scientists to deepen in the study of the 3D genome organization. Later on, authors highlight the role of epigenetic factors as DNA methylation and histone modifications in the regulation of the chromatin structure, which is essential to control every cellular process. Taking into account that these epigenetic regulators are often dysregulated in cancer, this work also discuss the emergence of novel epigenetic inhibitors that target enzymes involved in DNA methylation and histone modification in order to restore the chromatin state in different cancer subtypes.
Having said that, some inquiries and comments are addressed below to improve the quality of the manuscript and make it stand out with respect to other reviews in the field:
Major comments
The introduction should be rewritten, since authors do not display the link between the epigenetic factors involved in the regulation of chromatin organization and those which participate in the formation of chromatin loops (CTCF and cohesins).
The second paragraph of the introduction, which describes CTCF and cohesins’ functions, should be removed, since it does not provide any relevant information about the role of epigenetic mutations in the initiation and progression of cancer.
Recent publications are missing in some sections of the manuscript (i.e. In the “Techniques to study chromatin interactions” section, the most current publication is from 2013; in the “Histone phosphorylation” one, it is from 2011).
Apart from the explanation of the basic techniques developed to study chromatin interactions, recent advances in chromatin conformation capture techniques for unravelling 3D genome organizations and their application in cancer studies should be mentioned. In consequence, the figure 1 should be modified accordingly.
There are recent publications that go into detail about the role of epigenetic modifications in chromatin organization. It would be interesting to include the most recent advances in this field in the manuscript.
The table 1 is incomplete. I am aware of the difficulty of including all epigenetic regulators described up to now, but, at least, the most representative examples of all the types you mention on the section 3 should be added.
Authors should go in depth about the relationship between DNA methylation and cancer, since there are a lot of genes that are regulated by this epigenetic modification and contribute to cancer development. Some key experts in the field should be mentioned.
In the “Epimutations and Cancer” section, authors point out the link between epigenetic modifications and cancer, and some of the druggable targets associated with DNA methylation and histone modification. However, it is not clear the role of “insulators” in this context, since it is reported that mutations in these proteins are involved in cancer, but not in an epigenetic manner. Maybe they should be included in a separate section.
The table 2 must be updated with the novel inhibitors against histone modifiers that have been approved to be used in clinics during the last 5 years or those that are currently in different phases of clinical trials.
The futures perspectives should include new approaches that could complement the current anticancer therapies beyond the targeting of epigenetic regulators.
Minor comments
The line 31 should be rewritten: “chromatin forms high order structures called fibers” or “chromatin forms high order structures such as fibers, chromosomes, etc.”
In the line 40-41, “cohesion protein” should be replaced by “cohesins” or “cohesin complex”.
It remains unclear the meaning of this expression in lines 79-80: “under diluted conditions”. Perhaps you are referring to “at limiting dilution”.
Within the epigenetic factors, authors only mention the most representative histone modifications. It would be interesting to include others that have recently been discovered by mass spectrometry (i.e. citrunillation, ADP ribosylation, etc.) and are currently being characterized.
It would be advisable to include chaperones and ATP-dependent remodeling complexes, which are involved in nucleosome positioning, in the “Chromatin remodeling” section of the table 1, instead of only referring to some subunit members of these complexes. Furthermore, they should be also mentioned in the text.
In line 131, authors claim that CpG islands in gene promoters are usually unmethylated in normal cells, but this is only true in those genes that are transcriptionally active in each cell type.
The last sentence of the “DNA methylation” section should be rewritten.
In the “Histone phosphorylation” section, the fact that histones can also be phosphorylated in threonine residues is not indicated.
Some references are missing in the table 2.
The format of the tables should be reviewed.
For a better understanding, it would be appreciated to indicate the criteria that authors have used to select the specific epigenetic targets selected for the table 2 and the section 4 of the manuscript.
Spelling mistakes and the use of punctuation marks, acronyms and abbreviations through the text should be checked and corrected.
Author Response
Reviewer’s comments: In black
Response to comments: In blue
Reviewer 1
The review titled “Epimutations and their effect on chromatin organization: exciting avenues for cancer treatment” of Mohammad A. and Jha S. describes, firstly, the evolution of the techniques that has helped scientists to deepen in the study of the 3D genome organization. Later on, authors highlight the role of epigenetic factors as DNA methylation and histone modifications in the regulation of the chromatin structure, which is essential to control every cellular process. Taking into account that these epigenetic regulators are often dysregulated in cancer, this work also discuss the emergence of novel epigenetic inhibitors that target enzymes involved in DNA methylation and histone modification in order to restore the chromatin state in different cancer subtypes.
Having said that, some inquiries and comments are addressed below to improve the quality of the manuscript and make it stand out with respect to other reviews in the field:
Major comments
- The introduction should be rewritten, since authors do not display the link between the epigenetic factors involved in the regulation of chromatin organization and those which participate in the formation of chromatin loops (CTCF and cohesins).
We have amended the introduction section.
- The second paragraph of the introduction, which describes CTCF and cohesins’ functions, should be removed, since it does not provide any relevant information about the role of epigenetic mutations in the initiation and progression of cancer.
As suggested by the reviewer, the second paragraph that describes CTCF and cohesins is removed from the introduction paragraph and added to section 4 of the manuscript, where we discuss mutations in CTCF and cohesin in cancers.
- Recent publications are missing in some sections of the manuscript (i.e. In the “Techniques to study chromatin interactions” section, the most current publication is from 2013; in the “Histone phosphorylation” one, it is from 2011).
Apologies for the oversight; we have updated the reference in the “Techniques to study chromatin interactions” section.
- Apart from the explanation of the basic techniques developed to study chromatin interactions, recent advances in chromatin conformation capture techniques for unravelling 3D genome organizations and their application in cancer studies should be mentioned. In consequence, the figure 1 should be modified accordingly.
Thank you for the suggestion. In the revised manuscript, we have updated the text (lines 119-123) and figure 1 accordingly.
- There are recent publications that go into detail about the role of epigenetic modifications in chromatin organization. It would be interesting to include the most recent advances in this field in the manuscript.
Appreciate this suggestion by the reviewer. The modified manuscript now includes a discussion on the role of epigenetic modification in chromatin organization on line 241 onwards.
- The table 1 is incomplete. I am aware of the difficulty of including all epigenetic regulators described up to now, but, at least, the most representative examples of all the types you mention on the section 3 should be added.
We agree with the reviewer’s comments and have modified table 1, where we have now reorganized them as writers, readers, and erasers.
- Authors should go in depth about the relationship between DNA methylation and cancer, since there are a lot of genes that are regulated by this epigenetic modification and contribute to cancer development. Some key experts in the field should be mentioned.
We are thankful for this comment and have modified the section. Since multiple studies discuss the relationship between DNA methylation and cancer, we have briefly discussed them and cited relevant studies.
- In the “Epimutations and Cancer” section, authors point out the link between epigenetic modifications and cancer, and some of the druggable targets associated with DNA methylation and histone modification. However, it is not clear the role of “insulators” in this context, since it is reported that mutations in these proteins are involved in cancer, but not in an epigenetic manner. Maybe they should be included in a separate section.
As suggested by the reviewer, we have included a separate section 4.2, where we have discussed the role of the insulators in cancers
- The table 2 must be updated with the novel inhibitors against histone modifiers that have been approved to be used in clinics during the last 5 years or those that are currently in different phases of clinical trials.
Thank you for this suggestion. We have updated table 2 in the revised manuscript.
- The futures perspectives should include new approaches that could complement the current anticancer therapies beyond the targeting of epigenetic regulators.
As suggested by the reviewer, we have updated the future perspective.
Minor comments
- The line 31 should be rewritten: “chromatin forms high order structures called fibers” or “chromatin forms high order structures such as fibers, chromosomes, etc.”
As suggested, line 31 has been modified.
- In the line 40-41, “cohesion protein” should be replaced by “cohesins” or “cohesin complex”.
We agree with the suggestion and have changed it to “cohesin complex”.
- It remains unclear the meaning of this expression in lines 79-80: “under diluted conditions”. Perhaps you are referring to “at limiting dilution”.
Yes, we meant “under diluted conditions” and have amended it in the revised manuscript.
- Within the epigenetic factors, authors only mention the most representative histone modifications. It would be interesting to include others that have recently been discovered by mass spectrometry (i.e. citrunillation, ADP ribosylation, etc.) and are currently being characterized.
We agree with the suggestion and have now updated the histone modification section accordingly.
- It would be advisable to include chaperones and ATP-dependent remodeling complexes, which are involved in nucleosome positioning, in the “Chromatin remodeling” section of the table 1, instead of only referring to some subunit members of these complexes. Furthermore, they should be also mentioned in the text.
Thank you for this suggestion. We have revised the manuscript and excluded the remodeling complexes since we only focused on the epigenetic writers, readers, and erasers. We have updated table 1 accordingly.
- In line 131, authors claim that CpG islands in gene promoters are usually unmethylated in normal cells, but this is only true in those genes that are transcriptionally active in each cell type.
We agree with the reviewer’s comment and have amended it accordingly.
- The last sentence of the “DNA methylation” section should be rewritten.
As suggested by the reviewer, we have updated the “DNA methylation” section.
- In the “Histone phosphorylation” section, the fact that histones can also be phosphorylated in threonine residues is not indicated.
Apologies for the oversight. In the revised manuscript, we have included threonine in histones also to be phosphorylated.
- Some references are missing in the table 2.
We have added the missing references in table 2.
- The format of the tables should be reviewed.
Thanks for the comments; we have reorganized table 2 in the revised manuscript.
- For a better understanding, it would be appreciated to indicate the criteria that authors have used to select the specific epigenetic targets selected for the table 2 and the section 4 of the manuscript.
Thanks for the suggestion; we have now revised table 2 and section 4 and presented them as (4.1) writers, readers, and erasers and (4.2) Loops, TADs, and insulators.
- Spelling mistakes and the use of punctuation marks, acronyms and abbreviations through the text should be checked and corrected.
Apologies, we corrected these oversights.
Reviewer 2 Report
In the present review, Asad Mohammad and Sudhakar Jha have described techniques involved in studying the 3D organization of genomes, DNA methylation and histone modification, epigenetic deregulation in cancer, and epigenetic therapies targeting the tumor.
The review potentially has a broad interest in cancer epigenetics. However, there are some issues to be revised before considering publication in the Cancers.
Major concerns
1. The factors in the table1 are just regular epigenetic factors, not factors organizing 3D genome organization. Authors should revise table1 and include the factors involved in building-up TAD, their functions, types of mutation and associated cancers, and references in the table1.
2. Should describe the factors in detail in the section “3”.
3. Should elaborate on the Figure1 in terms of detailed technical procedures of TAD.
Author Response
Reviewer’s comments: In black
Response to comments: In blue
Reviewer 2
In the present review, Asad Mohammad and Sudhakar Jha have described techniques involved in studying the 3D organization of genomes, DNA methylation and histone modification, epigenetic deregulation in cancer, and epigenetic therapies targeting the tumor.
The review potentially has a broad interest in cancer epigenetics. However, there are some issues to be revised before considering publication in the Cancers.
Major concerns
- The factors in the table1 are just regular epigenetic factors, not factors organizing 3D genome organization. Authors should revise table1 and include the factors involved in building-up TAD, their functions, types of mutation and associated cancers, and references in the table1.
Thank you for the suggestion. In the revised manuscript, we have discussed the role of TAD in a separate section 4.2.
- Should describe the factors in detail in the section “3”.
We have revised the manuscript as suggested by the reviewer.
- Should elaborate on the Figure1 in terms of detailed technical procedures of TAD.
As suggested by the reviewer, we have discussed the formation of the loop, TAD, and high-order structures in a new section 4.2, and have also added a new figure 3.

Round 2
Reviewer 1 Report
As a reviewer, I really appreciate the work carried out by the authors to improve the quality of the manuscript, but there are still some issues that should be modified before publishing.
Major comments
Bibliography should be carefully reviewed. The total number of references cited in the text do not correspond to those included in the “References” section. Because of that, most of them seem to be positioned incorrectly along the manuscript.
I have seen the improvements performed in the section 2 and the figure 1 of the manuscript. However, other chromatin conformation capture techniques are not mentioned in the text, but they are reported in scientific articles that authors include in the “References” section. Both text and figure 1 should be amended accordingly.
The table 1 remains incomplete. The new classification of epigenetic factors as writers, readers or erasers is highly appreciated, but the proteins/families included are not still enough to have a comprehensive overview of these factors. Furthermore, some epigenetic modifiers authors mentioned in the section 3 and the table 2 (as new druggable targets) have not been added to table 1.
Minor comments
The line 31 should be rewritten: “chromatin forms high order structures such as fibers, chromosomes, etc.”
In the line 40-41, “cohesin complex proteins” should be replaced by “the cohesin complex”.
Also in line 41, it would be advisable to substitute “The epigenetic process” by “Epigenetics”.
The meaning of the expression “under dilute conditions” in the line 71 is still unclear and it could be removed to facilitate the understanding due to the fact that it is not relevant for the description of the technique.
Authors mention “microarray or sequencing” in the figure 1.c, which are referring to the section 2.2 of the manuscript, but there are no references to microarrays in the text.
In lines 144-145, the expression “while in normal tissues, gene promoters are typically unmethylated” should be replaced by “while in normal tissues, these CpG islands are typically unmethylated”.
The second sentence of the “ADP-ribosylation” section should be rewritten for its better understanding.
It would be advisable to substitute “Various pathological events” by “Some pathological events” in line 204.
It would be needed to add a brief section (3.9) to mention other minor histone modifications already reported in the bibliography.
The word “cohesion” should be replaced by “cohesin” along the text.
The name of the genes should be written in italics.
In lines 316-317, some information is missing. Maybe authors wanted to say that: “bring upstream regulatory elements in close proximity to gene promoters”.
The sentence placed in the lines 364-366 should be rewritten.
In lines 385-387, authors mention an inactivating mutation in the “cohesion complex gene”, but taking into account that the cohesin complex has different subunits, it is unclear which gene harbours the mutation.
It is highly recommended not to abbreviate the words when acronyms are first used. It should be reviewed along the text.
Spelling mistakes and the use of punctuation marks through the text should be checked and corrected again.
Author Response
Reviewer’s comment: In Black
Response to comments: In Blue
Reviewer 1
Comments and Suggestions for Authors
As a reviewer, I really appreciate the work carried out by the authors to improve the quality of the manuscript, but there are still some issues that should be modified before publishing.
Major comments
Bibliography should be carefully reviewed. The total number of references cited in the text do not correspond to those included in the “References” section. Because of that, most of them seem to be positioned incorrectly along the manuscript.
Apologies for the oversight; we have carefully reviewed and cited the bibliography correctly.
I have seen the improvements performed in the section 2 and the figure 1 of the manuscript. However, other chromatin conformation capture techniques are not mentioned in the text, but they are reported in scientific articles that authors include in the “References” section. Both text and figure 1 should be amended accordingly.
We agree with the reviewer’s comment and have updated the text and figure 1 in section 2.
The table 1 remains incomplete. The new classification of epigenetic factors as writers, readers or erasers is highly appreciated, but the proteins/families included are not still enough to have a comprehensive overview of these factors. Furthermore, some epigenetic modifiers authors mentioned in the section 3 and the table 2 (as new druggable targets) have not been added to table 1.
Thank you for the suggestion; we have updated the epigenetic factors in table 1.
Minor comments
The line 31 should be rewritten: “chromatin forms high order structures such as fibers, chromosomes, etc.”
As suggested by the reviewer, we have modified the lines accordingly.
In the line 40-41, “cohesin complex proteins” should be replaced by “the cohesin complex”.
We have replaced “cohesin complex proteins” with “the cohesin complex.”
Also, in line 41, it would be advisable to substitute “The epigenetic process” by “Epigenetics”.
Line 41 is amended accordingly.
The meaning of the expression “under dilute conditions” in the line 71 is still unclear and it could be removed to facilitate the understanding due to the fact that it is not relevant for the description of the technique.
As suggested by the reviewer, we amended the sentence “under dilute conditions” (line 71) for better understanding.
Authors mention “microarray or sequencing” in the figure 1.c, which are referring to the section 2.2 of the manuscript, but there are no references to microarrays in the text.
Apologies for the inconsistency in figure 1. C, and the text in section 2.2. Both the text and figure are updated in the revised manuscript.
In lines 144-145, the expression “while in normal tissues, gene promoters are typically unmethylated” should be replaced by “while in normal tissues, these CpG islands are typically unmethylated”.
As the reviewers suggested, we modified lines 144-145 by “while in normal tissues, these CpG islands are typically unmethylated.”
The second sentence of the “ADP-ribosylation” section should be rewritten for its better understanding.
The second sentence in section 3.8 (ADP-ribosylation) is rewritten.
It would be advisable to substitute “Various pathological events” by “Some pathological events” in line 204.
As suggested by the reviewers, we have updated “Various pathological events” with “Some pathological events” in line 204.
It would be needed to add a brief section (3.9) to mention other minor histone modifications already reported in the bibliography.
Thank you for the suggestion, in section 3 we have mentioned other histone modifications, such as proline isomerization and histone glycosylation.
The word “cohesion” should be replaced by “cohesin” along the text.
We have replaced “cohesion” with “cohesin” throughout the manuscript.
The name of the genes should be written in italics.
We have reviewed the manuscript and amended the gene name in italics.
In lines 316-317, some information is missing. Maybe authors wanted to say that: “bring upstream regulatory elements in close proximity to gene promoters”.
Thank you for the suggestion; we have corrected the sentence (lines 365-366).
The sentence placed in the lines 364-366 should be rewritten.
As suggested by the reviewers, we have amended the sentence (lines 415-417).
In lines 385-387, authors mention an inactivating mutation in the “cohesion complex gene”, but taking into account that the cohesin complex has different subunits, it is unclear which gene harbours the mutation.
Thank you for the comment. In AML, there are mutations in cohesin complex genes (STAG1, STAG2, SMC1A, SMC3, and RAD21) https://doi.org/10.1182/blood-2013-07-518746. To make it simple, we removed the line in the revised manuscript.
It is highly recommended not to abbreviate the words when acronyms are first used. It should be reviewed along the text.
As suggested by the reviewer, we reviewed the manuscript and amended it accordingly.
Spelling mistakes and the use of punctuation marks through the text should be checked and corrected again.
We have corrected the oversights.
Reviewer 2 Report
I have no further concern(s) on the revision
Author Response
Reviewer’s comment: In Black
Response to comments: In Blue
Reviewer 2
Comments and Suggestions for Authors
I have no further concern(s) on the revision
We are glad that the revised manuscript has addressed all the reviewer's concerns and appreciate his/her feedback to improve the draft.

Round 3
Reviewer 1 Report
As I mentioned in my previous review, I appreciate so much authors’ modifications and comments, but there are still some aspects that have to be amended:
Major comment
Bibliography should be carefully reviewed AGAIN. I have seen the changes authors have carried out throughout the text, but:
1. There is still an inconsistency between the total number of references cited (178 is the highest number that appears in the text) and those included in the “References” section (123).
2. Some references are missing.
3. Several references are not correctly cited in order of appearance in the text.
4. The “References” section of the manuscript has not been modified since its first version.
To summarize, the bibliography of this article needs some strong improvement. If it is not modified, I would recommend that the manuscript is not accepted for publication.
My rationale for taking this decision is that I consider that bibliography is a core aspect of a scientific article as it defines the ground for the current investigation but also support fellow colleagues to obtain more information about this research field.
Minor comments
These comments are only focused on improving the quality of the manuscript in its current version and arise from reading the text repeatedly:
The first sentence in the “Summary” section should be written in plural: “Epigenetic modifications, such as DNA methylation and histone modification, have…”.
The line 31 should be rewritten: “chromatin forms high-order structures such as fibers, loops, domains, etc.”
In lines 41 and 346, “cohesin complex” should be replaced by “THE cohesin complex”.
Spelling mistakes in figure 1 should be checked and corrected. Furthermore, the bottom part of the figure is not fully readable.
Authors should rewrite the sentences located in lines 72-75, since the modifications performed in them are not well understood.
In the line 90, authors should introduce a hyphen after the word “gene” to facilitate the understanding: the gene- or regulatory interaction.
Authors should be consistent with the use of HAT or HATs, and HDAC or HDACs along the manuscript.
In the line 140, the expression “increase in high throughput” should be replaced by “increase in throughput”, since otherwise, it sounds a little redundant.
In the line 290, the expression “such as methylation” is inconsistent with the meaning of the sentence. For instance, it could be replaced by “such as those that are methylated”.
Following my advice, authors had included new examples of writers and readers in table 1 (maybe they could also add other PRMTs mentioned in table 2, other BRD members and the BET proteins). They seem to be mainly interested in histone acetylation/deacetylation and methylation/demethylation, and because of that, do not show any references related to other histone modifications in tables 1 and 2. If it is correct, this should be mentioned in both table footnotes, the figure 2 and the section 4 of the manuscript.
The format of both tables 1 and 2 is different. Moreover, the table 2 should be amended, since it is not clear if some histone modifiers’ families are writers, readers or erasers.
The gene “BRCA1” in the line 251 should be written in italics.
In figure 2, the acronym of 5-methylcytosine should be modified: 5-mC (C in capital letters).
The acronym “CRC” in the line 388 is referred to colorectal cancer and this should be indicated in the text for a better understanding.
The expression “gene expressions” in the line 493 should be replaced by “gene expression”.
The text format should be checked and corrected, as some parts seem to have different typos and font sizes.
Author Response
Reviewer’s comment: In Black
Response to comments: In Blue
Comments and Suggestions for Authors
As I mentioned in my previous review, I appreciate so much authors’ modifications and comments, but there are still some aspects that have to be amended:
Major comment
Bibliography should be carefully reviewed AGAIN. I have seen the changes authors have carried out throughout the text, but:
- There is still an inconsistency between the total number of references cited (178 is the highest number that appears in the text) and those included in the “References” section (123).
- Some references are missing.
- Several references are not correctly cited in order of appearance in the text.
- The “References” section of the manuscript has not been modified since its first version.
To summarize, the bibliography of this article needs some strong improvement. If it is not modified, I would recommend that the manuscript is not accepted for publication.
My rationale for taking this decision is that I consider that bibliography is a core aspect of a scientific article as it defines the ground for the current investigation but also support fellow colleagues to obtain more information about this research field.
Our sincere apologies for the errors in reference citated; while reviewing the documents we realized that the error in bibliography was when we created the documents with track changes before submission. We have carefully reviewed and cited the bibliography with correct number.
Minor comments
These comments are only focused on improving the quality of the manuscript in its current version and arise from reading the text repeatedly:
The first sentence in the “Summary” section should be written in plural: “Epigenetic modifications, such as DNA methylation and histone modification, have…”.
Thank you for the suggestion; we have written the sentence in plural form.
The line 31 should be rewritten: “chromatin forms HIGH-ORDER STRUCTURES such as fibers, loops, domains, etc.”
As suggested by the reviewer, we have rewritten the sentence in the line 31-32.
In lines 41 and 346, “cohesin complex” should be replaced by “THE cohesin complex”.
We have replaced “cohesin complex” with “the cohesin complex.”
Spelling mistakes in figure 1 should be checked and corrected. Furthermore, the bottom part of the figure is not fully readable.
Apologies for the spelling error, we have replaced the figure 1 accordingly.
Authors should rewrite the sentences located in lines 72-75, since the modifications performed in them are not well understood.
As suggested by the reviewer, we have modified the sentences for better understanding.
In the line 90, authors should introduce a hyphen after the word “gene” to facilitate the understanding: the gene- or regulatory interaction.
Thank you for the suggestion, we have updated accordingly.
Authors should be consistent with the use of HAT or HATs, and HDAC or HDACs along the manuscript.
As suggested by the reviewer, we have reviewed “HAT or HATs”, and “HDAC or HDACs” and used when referring to the domain or family in the manuscript.
In the line 140, the expression “increase in high throughput” should be replaced by “increase in throughput”, since otherwise, it sounds a little redundant.
Thank you for the suggestion, we have replaced “increase in high throughput” with “increase in throughput”.
In the line 290, the expression “such as methylation” is inconsistent with the meaning of the sentence. For instance, it could be replaced by “such as those that are methylated”.
We agree with the reviewer’s comment and modified with “such as those that are methylated”.
Following my advice, authors had included new examples of writers and readers in table 1 (maybe they could also add other PRMTs mentioned in table 2, other BRD members and the BET proteins). They seem to be mainly interested in histone acetylation/deacetylation and methylation/demethylation, and because of that, do not show any references related to other histone modifications in tables 1 and 2. If it is correct, this should be mentioned in both table footnotes, the figure 2 and the section 4 of the manuscript.
As suggested by the reviewer, we have updated the table 1 and modified accordingly.
The format of both tables 1 and 2 is different. Moreover, the table 2 should be amended, since it is not clear if some histone modifiers’ families are writers, readers or erasers.
As suggested by the reviewer, we have carefully reviewed and updated the table 2. For clarity, we have color coded the tables as per the families.
The gene “BRCA1” in the line 251 should be written in italics.
Thank you for the suggestion; in the revised manuscript “BRCA1” is in italics.
In figure 2, the acronym of 5-methylcytosine should be modified: 5-mC (C in capital letters).
Apologies for the mistake in acronym, we have updated 5-mc with 5-mC in figure2.
The acronym “CRC” in the line 388 is referred to colorectal cancer and this should be indicated in the text for a better understanding.
Thank you for the suggestion; we have updated colorectal cancer.
The expression “gene expressions” in the line 493 should be replaced by “gene expression”.
We have replaced “gene expressions” by “gene expression”.
The text format should be checked and corrected, as some parts seem to have different typos and font sizes.
As suggested by the reviewer, we have corrected the oversights.
